# UV-Activated, Transparent Oxygen Scavenger Coating Based on Inorganic–Organic Hybrid Polymer (ORMOCER®) with High Oxygen Absorption Capacity

Sabine Amberg-Schwab [1,†], Kajetan Müller [2,3], Ferdinand Somorowsky [1,*] and Sven Sängerlaub [2,4,†]

[1]   Fraunhofer-Institute for Silicate Research ISC, Neunerplatz 2, 97082 Wuerzburg, Germany
[2]   Fraunhofer Institute for Process Engineering and Packaging IVV, Giggenhauser Strasse 35, 85354 Freising, Germany
[3]   Faculty of Mechanical Engineering, University of Applied Science Kempten, Bahnhofstraße 61, 87435 Kempten, Germany
[4]   TUM School of Life Sciences Weihenstephan, Chair of Food Packaging Technology, Technical University of Munich, Weihenstephaner Steig 22, 85354 Freising, Germany
*   Correspondence: ferdinand.somorowsky@isc.fraunhofer.de
†   These authors contributed equally to this work.

**Abstract:** Oxygen scavengers are used to reduce the oxygen permeation of packaging (active barrier) and to absorb oxygen from its direct environment, e.g., a headspace of packaged food. Few oxygen scavenger coatings have been developed. Therefore, in this study, a novel oxygen scavenger coating has been developed. It is based on inorganic–organic polymers (ORMOCER®). The oxygen absorption reaction is activated by UV light. The scavenger was synthesized, coated on aluminum foil, subsequently dried and afterwards laminated with a polyethylene sealing layer. UV light activates the oxygen scavenging reaction. The oxygen absorption capacity, measured at 23 °C and 0% r.h., was $242 \pm 8$ mg oxygen/g scavenger coating. When the oxygen scavenger coating layer was laminated by using a two-component polyurethane laminating adhesive, the absorption capacity was hardly reduced, with a measured absorption capacity of $223 \pm 18$ mg oxygen/g scavenger coating. In an experimental packaging sample with the oxygen scavenger coating with a thickness (dry) of 3 μm and 18 μm, near-zero mbar oxygen partial pressure was reached by the non-laminated oxygen scavenger coatings within two days, and within about 20 days when laminated with a polyurethane laminating adhesive and a PE-layer on the oxygen scavenger layer. The oxygen partial pressure was kept near zero mbar for 500 days, whereas in the experimental packaging without oxygen scavenger, the oxygen partial pressure increased to 110 mbar during this time. The developed oxygen scavenger based on inorganic–organic polymers can be applied as wet chemical coating on various surfaces with standard application procedures. Application scenarios are oxygen-sensitive goods such as food, pharmaceutical products and cosmetics.

**Keywords:** inorganic–organic polymers; hybrid polymers; barrier coatings; ORMOCER®; active barrier layers; oxygen scavengers; barrier laminates; active packaging

## 1. Introduction

### 1.1. Motivation for Using Oxygen Scavengers in Packaging

A wide range of products, such as foods, are susceptible to oxidative damages. For foods, oxygen can cause changes in flavor, taste and color, nutrient loss and aerobic microbial growth, rancidity or vitamin loss. Examples of sensitive foods are various kinds of sausages, milk and coffee cream, nuts, beer, edible oil, wine and juices [1–12].

Oxygen-sensitive foods spoil when reacting with just 1–200 ppm (mg/kg food) oxygen [13,14]. The levels of residual oxygen in packaging systems can be higher because of several reasons: solved oxygen in the food matrix, oxygen in pores of the food and

residual oxygen after gas flushing (between 0.1 to 2 vol.%). It is worth mentioning that fats and oils solve up to five or six times more oxygen than water [15]. Therefore, not only is the prevention of oxygen access to the product an important task in packaging, but so is the absorption of oxygen already present at the time of packaging. Important access paths for oxygen to filled goods are the inherent permeability of the packaging material, and defects in the package. For very sensitive packaged goods, e.g., pharmaceuticals, drugs and cosmetics, and for many technical applications, the passive barrier properties—also in combination with inert gas flushing—are insufficient [16].

Suitable materials which can be used for these sensitive areas in food packaging combine passive barrier layers with active barrier layers, e.g., oxygen-consuming layers ('oxygen scavengers') [17–20]. A strategy to incorporate oxygen scavengers into packaging materials is blending the monolayer packaging material with an oxygen scavenger, e.g., PET bottles with oxygen scavengers [21–25], or the integration of a separate layer of a multilayer film structure [26–31]. There, the oxygen scavengers act as an active barrier and/or they absorb oxygen from the headspace surrounding the packaged food.

### 1.2. Oxygen Scavenger Coating Based on Inorganic–Organic Polymers (ORMOCER®)

The author of this study, Sabine Amberg-Schwab, has developed an oxygen scavenger based on inorganic–organic polymers [32–35]. With a new kind of barrier coating material, namely ORMOCER® (Figure 1), it is possible to obtain a coating that absorbs oxygen.

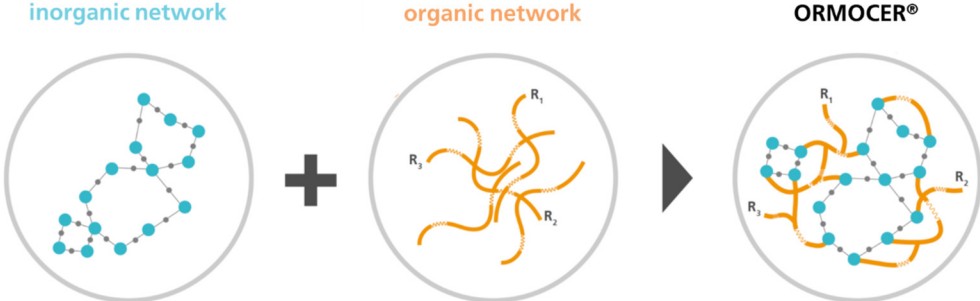

**Figure 1.** Hybrid polymers (ORMOCER® developed by Fraunhofer ISC, trademark of Fraunhofer Gesellschaft zur Förderung der angewandten Forschung e.V., München) [Fraunhofer ISC].

ORMOCER®-based coatings are hybrid polymers that can be synthesized by the sol–gel technique. If extremely low permeation values are needed, the combination of hybrid polymer coatings with thin inorganic layers ($SiO_x$, $AlO_x$) is very effective and leads to permeation values for oxygen and water vapor below $10^{-3}$ $cm^3/(m^2$ d bar) or $g/(m^2$ d), 85% r.h. to 0% r.h. at 23 °C [16]. These passive barrier layers can be further improved by applying active oxygen barrier layers, which have been developed for the food packaging industry [32,36]. This approach makes these multilayer laminates promising candidates for special applications in the food packaging industry as well as for sophisticated applications in technical areas: the encapsulation of sensitive organic devices such as solar cells, organic light-emitting diodes, or polymer electronic systems.

The functional principle of the newly developed oxygen scavenger system is based on a photo-initiated, metal-catalyzed oxidation of a cyclo-olefin bonded chemically to a silicate backbone. This concept permits the activation of the scavenging process by UV light and prevents the formation of low-molecular oxidation products that may decrease the quality of the packaged goods or may be toxic. For packaging-related applications, UV activation is carried out directly before the packaging process. A similar strategy was followed by Chevron Chemical Company and Cryovac Inc., a subsidiary of Sealed Air Corporation [20,37]. In their patents, they describe the production of a terpolymer with oxygen scavenging properties via partial transesterification of an ethylene/methacrylate copolymer with an alcohol comprising a cyclic olefin compound by means of reactive extrusion. The major difference between the Chevron concept and the system described



here is the backbone to which the cyclic olefin groups are linked. While the ethylenic backbone of the Chevron scavenger system is pre-formed and subsequently modified to introduce functional groups, the hybrid polymer-based system is formed in a one-step synthesis by hydrolysis of the corresponding alkoxysilanes (Figure 2). In addition, the resulting sol can be applied to any suitable film substrate by common lacquer coating techniques. It forms a transparent layer. Usual layer thicknesses can be between 2 and 20 μm, according to the experimental experience of the authors.

**Figure 2.** Proposed mechanism of the transition metal-catalyzed oxidation of the silylated cyclic olefin compound [Fraunhofer ISC].

The active oxygen scavenger molecule (cyclo-olefin) is covalently fixed to the silicate network of the hybrid polymer through a condensation reaction. This concept of oxygen scavenging permits the activation of the scavenging process by UV light and prevents the formation of low-molecular oxidation products that may decrease the quality of the packaged goods or may be toxic (Figure 2). ORMOCER$^\circledR$ coatings are generally characterized by complete crosslinking and a very dense network. As a result, any cleavage products that may occur can be prevented from migrating, and no monomers migrate out of the layer.

After the application of the oxygen scavenger lacquer layer, the activation is effected by UV light with a UV intensity of 6 to 7 J/cm$^2$. The reaction mechanism of 2-cyclohexenyltriethoxysilane shown in Figure 2 was confirmed by spectroscopic analysis of the oxidation products of an oxygen-scavenging layer [37]. Due to the activation of the oxygen scavenger layer by means of UV light, the oxygen scavenger capacity of the coating is independent of the humidity of the environment.

### 1.3. Intention of This Study

In previous studies of Fraunhofer ISC, different cobalt salts and concentrations were investigated for their catalytic effectivity. Cobalt(II) oleate turned out to be the most effective catalyst among the cobalt salts investigated, presumably due to its high mobility in the matrix. With the change from cobalt(II) 2-ethylhexanoate to cobalt(II) oleate, a significant increase in the capacity of the above-mentioned alkyl-modified hybrid polymer matrix could be achieved with a simultaneous increase in the oxidation rate. The result is an oxidation rate of 90% and an oxygen capacity of 58 cm$^3$/g layer within just three days of activation. Furthermore, the relationship between the network density and oxygen capacity of the matrix was examined using the parameters 'curing time' and 'type of solvent'. With a shorter curing time and high-boiling solvents, a higher degree of oxidation and,

thus, an increase in capacity could be achieved. The combination of this alkyl-modified hybrid matrix and the oxidation catalyst cobalt(II) oleate thus represents a UV-activatable scavenger system that offers high capacity with extremely fast kinetics and, in addition, high transparency and good adhesion properties of the layers on PET.

The objective of this study was to characterize our improved, flexible and transparent oxygen scavenger coatings in an experimental packaging sample with respect to their oxygen scavenging capacities. The secondary objective was to test different photo-initiators available on the market.

## 2. Materials and Methods

### 2.1. Preparation of Coating Sol OS 12

For the development of the oxygen scavenger systems, an active organosiloxane, which reacts with oxygen, was selected as the main component: 2-cyclohexenyltriethoxysilane (Evonik Industries AG, Essen, Germany).

To prepare a coating system based on this silane, it was combined with a hybrid polymer matrix consisting of an epoxy-modified silane (Evonik Industries AG, Essen, Germany) and/or a tetraethoxysilane (Evonik Industries AG, Essen, Germany) and metal alcoholates (Evonik Industries AG, Essen, Germany).

For the synthesis of the matrix system, a solution of 2-cyclohexenylethyltriethoxysilane and n-octyltriethoxysilane in 2-butoxyethanol was stirred and cooled in an ice bath. After acid hydrolysis (1N HCl) with 50 wt.% of the stoichiometric amount of $H_2O$ with respect to the hydrolyzable alkoxy groups, the mixture was stirred at room temperature for 2 h. As a second component, a solution prepared by mixing Zr-n-propoxide and ethyl acetoacetate was added. The completion of the hydrolysis reaction was controlled using Raman spectroscopy. (FT-Raman spectroscopic investigations were carried out on a c model RFS 100 (Bruker Corporation, Billerica, MA, USA), laser power on the samples: 500 mW, spectral resolution 4 cm$^{-1}$, with a Nd-YAG laser as excitation light source with an emission wavelength of 1064 nm [32].) To this coating matrix, Co(II)-oleate (<2 wt.%, catalyst; The Shepherd Chemical Company, Norwood, OH, USA), 1 wt.% of the photo-initiator Lucirin TPO (Ciba, now BASF, Ludwigshafen, Germany) and 1–3 wt.% antioxidant (Irganox 1010; product number 0304865, Ciba Specialty Chemicals Corporation, Basel, Switzerland) were added. The cobalt di-oleate was dissolved by means of an ultrasonic bath for 2 h at 23 °C. Afterwards, the bottle was stirred intensively for 5 min. After this procedure, the lacquer was ready to use. The curing of the layer can be performed between 80 °C and 130 °C for 1 h.

To identify if a specific photo-initiator influences the oxygen capacity and reaction kinetics of the oxygen scavenger layer, different photo-initiators were tested; OS 12a (Irgacure 184, Ciba, now BASF, Ludwigshafen, Germany), OS 12b (Lucirin TPO, Ciba, now BASF, Ludwigshafen, Germany) and OS 12c (Irgacure 500, Ciba, now BASF, Ludwigshafen, Germany) (OS: short for oxygen scavenger).

### 2.2. Coating of OS-Sol on Substrate

The scavenger coating was applied on aluminum foil with 100 μm thickness (substrate 1), on an Al-vapor-deposited PET (50 μm) film (PETmet, substrate 2) from Fraunhofer ISC as well as on a laminate consisting of PET (12 μm) and an Al (9 μm) foil (PET//adhesive//Al; substrate 3). All substrates had a format of A4. The oxygen scavenger lacquer was applied with a pipette in front of a profile rod. The profile rod was clamped in a semi-automatic coating device (K303 Control Coater Modell 625, Erichsen GmbH & Co., KG, Hemer, Germany). Then, the coating was pulled smooth with a coating velocity of 2 m/min. The profile rods were chosen for a wet film thickness of 12 and 60 μm.

The contact angle to water was between 64° and 68°.

The thickness of the oxygen absorbing layer after drying was around 2.3 μm, 3.2 ± 1.3 μm and 18.2 ± 3.5 μm. The curing was carried out in a drying oven (T6120, Heraeus, Hanau, Germany) at 160 °C for 0.5 and 5 to 10 min.

The addition of the oxygen scavenger function did not change the thermal and mechanical properties of the coated and laminated layers. The layers have sufficient thermal and mechanical stability for all processing and environmental conditions that occur in the packaging sector.

### 2.3. Multilayer Production

#### 2.3.1. Adhesive Preparation

As adhesion promoter, a 2-component polyurethane laminating adhesive that is often used in the packaging industry was selected. The adhesive was prepared from 100 g adhesion promoter (Liofol UK 3640, Henkel, Düsseldorf, Germany), 2 g of a hardener (Liofol UK 6800, Henkel, Düsseldorf, Germany) and 145 g ethyl acetate (Th. Geyer, Renningen, Germany). These materials were placed in a beaker. With a magnetic stirrer, the ingredients were homogenized at 23 °C for 5 min.

#### 2.3.2. Laminating the Sealing Layer

The adhesive was coated on the oxygen scavenger (OS) layer. The primer was placed in front of the profile rod with a pipette. Subsequently, the adhesive on the oxygen scavenger layer was pulled smooth. The wet layer thickness was around 12 μm. The adhesive layer was subsequently pre-dried with a hair dryer for 2 min and then stored in a drying oven at 80 °C for 5 min. The resulting dry film thickness was about 2 to 4 μm. Subsequently, by hand, PE-LD (INVOS, spol. s r.o., Březolupy, Czech Republic) film was pressed with a rubber roller onto the adhesion promoter layer. The LD-PE layer thickness was 30 μm.

### 2.4. UV Activation of Oxygen Scavenger

By means of UV lamps, the coated samples were activated. The activation of the oxygen scavenger systems were carried out on a Beltron 22/III with a UV performance of 8.2 J/cm$^2$ (2 UV lamps, each 1200 W, throughput speed of 0.4 m/min and exposure time of 30 s), see [32].

In addition, the samples were activated with another UV lamp (H radiator, UVC 200–280 nm, Dr. Höhnle AG, Graefelfing, Germany; at a coating pilot plant of the Fraunhofer IVV, take-off speed 1 m/min). The UV intensity was 6–7 J/cm$^2$.

Immediately after activation, within 1 to 2 min, the samples were introduced into measuring cells and the oxygen absorption measurements started.

### 2.5. Determination of Oxygen Scavenger Reactivity

Measuring cells with an internal headspace volume of 150 cm$^3$ were used for determining the oxygen absorption capacity. Therein, 250 cm$^2$ of oxygen-absorbing film were incorporated. The initial oxygen concentration was 20 vol.%. The temperature was 23 °C. The oxygen partial pressures in the cells were measured with an oxygen electrode (WTW Clark-Elektrode CellOx 325 equipped with a measuring device Oxi 315i, Weilheim, Germany) or with an optical system (PreSens, Regensburg, Germany). From the reduction in the oxygen partial pressure, the consumed amount of oxygen and, thus, the oxygen absorption capacity of the oxygen scavenger was calculated. The r.h. during sample preparation was circa 50%.

### 2.6. Experimental Packaging

The aim of the experimental packaging was to simulate the conditions of a defective snack pack: headspace volume 118 cm$^3$, area inside the snack pack (oxygen absorbing area) 250 cm$^2$ and oxygen permeability of a capillary with a diameter of 10 microns as described before [38].

The measuring cells used had an internal volume of 150 cm$^3$. Into the cells, 80 g of glass beads were inserted to adjust the headspace volume to 118 cm$^3$. A sample of 250 cm$^2$ of film was stored in the measuring cells. The measuring cells were sealed with a 50 μm thick PET film. This set a defined permeability of 0.03 cm$^3$ O$_2$/day, which corresponds to

the oxygen permeability of a pore, with a diameter of 10 μm, of the snack pack. The initial oxygen concentration was 2 vol.%. This corresponds to an oxygen partial pressure of about 20 mbar. Overall, ca. 10 mg oxygen will be present in a pack resulting in a shelf life of 180 days, calculated from the oxygen permeation and the residual oxygen in the headspace. The samples were stored at 23 °C. For each sample, a 5-fold determination was applied. In Figure 3, an experimental packaging is shown.

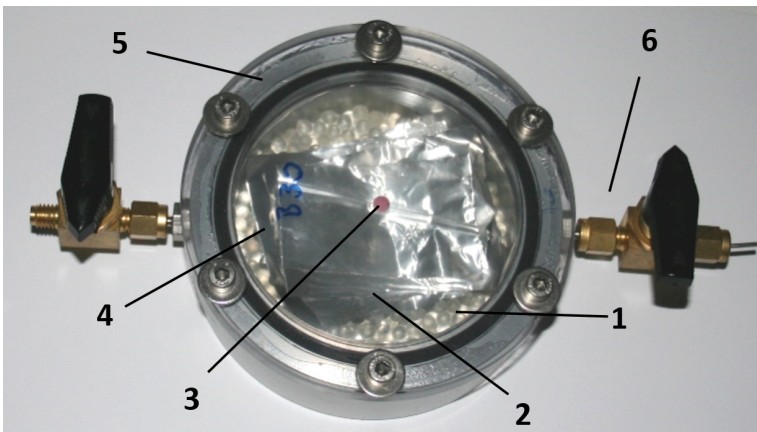

**Figure 3.** Experimental packaging [Fraunhofer IVV], 1: glass beads to adjust headspace volume, 2: film with oxygen scavenger, 3: oxygen sensor (from PreSens), 4: PET film, thickness 50 μm with defined oxygen transmission rate, 5: clamping ring, 6: valves to set the initial oxygen concentration.

## 3. Results and Discussions

### 3.1. Oxygen Absorption

#### 3.1.1. Oxygen Absorption Capacity

At the beginning of the measurements, the oxygen concentration in the measuring cell was 20 vol.%.

In Figure 4, the oxygen absorption of OS 12 with different photo-initiators is shown. The photo-initiators used were the photo-initiator Irgacure 184 for OS 12a, the photo-initiator Lucirin TPO for OS 12b and the photo-initiator Irgacure 500 for OS 12c. The fastest absorption rate was exhibited by OS 12b. After just five days, the oxygen absorption capacity was almost exhausted. System OS 12a consumed the least amount of oxygen during this period, compared to OS 12b. However, after 25 days, system OS 12c achieved the same high oxygen capacity as system OS 12b, while system OS 12a was still slightly lower. Thus, three systems with different reaction kinetics were available. The differences between the systems were achieved by the use of different photo-initiators and the degree of crosslinking of the inorganic and organic networks formed in the layers. This enables adapted kinetics for oxygen removal for different packaging purposes.

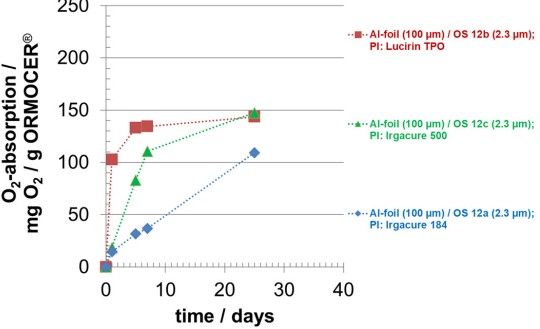

**Figure 4.** Oxygen absorption of oxygen scavenger (OS) 'ORMOCER® OS12' with different photo-initiators (PI); curing time: 0.5 min; OS coating thickness (dry) around 2.3 μm.

In Figures 5 and 6, the oxygen absorption of OS12 with and without sealing layer PE is shown. Without a sealing layer, the oxygen capacity of the two systems is almost exhausted after five days. The reaction kinetics of the two systems is comparable here. The system OS 12b (Lucrin TPO) reacts as quickly (Figure 5) as depicted in Figure 4. However, the absorption capacity was 200 mg oxygen/g scavenger and, therefore, higher than the result of Figure 4 by 60 mg oxygen/g scavenger. The reason may be premature reaction during preparation in the case of Figure 4. OS 12a with Irgacure 184 reacted faster (Figure 6) than at the first trial (Figure 4). We assume that the catalyst and photo-initiator was better distributed in the second case. In a previous version of this oxygen scavenger system, the absorption rate was similar but the absorption capacity was lower (90 mg oxygen/g scavenger) [32].

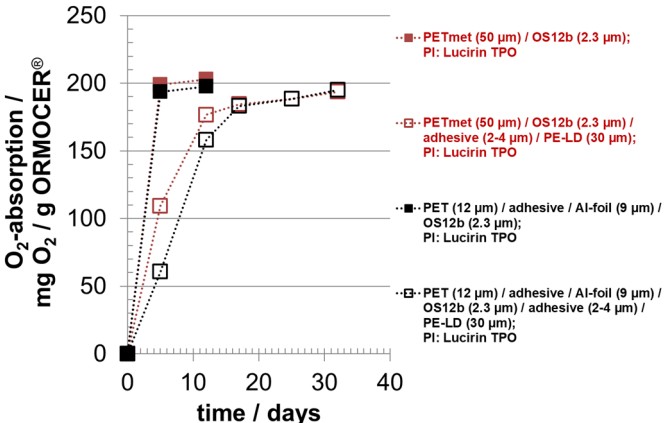

**Figure 5.** Oxygen absorption of oxygen scavenger 'ORMOCER® OS12b' with photo-initiator (PI) Lucirin TPO; curing time: 5 min; OS coating thickness (dry) around 2.3 µm; with and without sealing layer PE-LD.

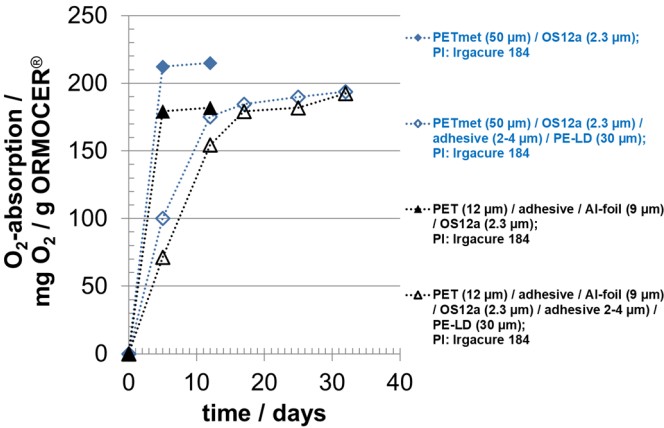

**Figure 6.** Oxygen absorption of oxygen scavenger 'ORMOCER® OS12b' with photo-initiator (PI) Irgacure 184; curing time: 5 min; OS coating thickness (dry) around 2.3 µm; with and without sealing layer PE-LD.

By lamination, the absorption rate was reduced approximately by a factor of 3. A suitable hypothesis is that the adhesive acts as barrier for oxygen. PUR adhesive has an oxygen permeability of around 10 cm$^3$ O$_2$ 100 µm (STP)/(m$^2$ d bar) at 23 °C and 50 r.h. and is, therefore, a medium oxygen barrier material [39–41]. The adhesive between the oxygen scavenger and the PE-LD film had no effect or a negligible effect on the oxygen absorption capacity. This result is important because, practically without losing oxygen absorption capacity, the possible migration of photo-initiators from the oxygen scavenger layer to a packaged good could be prevented by using an adhesive-laminated functional

barrier layer, instead of a PE-LD film, as in the experiments of this study. Photo-initiators are seen critically for packaging of food products.

The system 12b with Lucirin TPO was selected again for the analysis of the oxygen absorption, this time as five-fold determination. In Figure 7a,b, the oxygen concentration and in Figure 7c, the oxygen absorption, calculated from the reduction in the oxygen concentration, are shown. Almost all oxygen was scavenged in the measuring cells, i.e., the oxygen scavenger also binds oxygen at low oxygen concentration. The oxygen scavenger based on the ORMOCER® absorbs 240 mg oxygen/g scavenger without a sealing layer. With a sealing layer, the oxygen absorption capacity was slightly reduced to 220 mg oxygen/g scavenger. This result shows that the adhesive had no significant effect on the oxygen scavenger. These measured values are 50 to 100 mg oxygen/g scavenger lower than those in the previous trials presented in Figures 4 and 5.

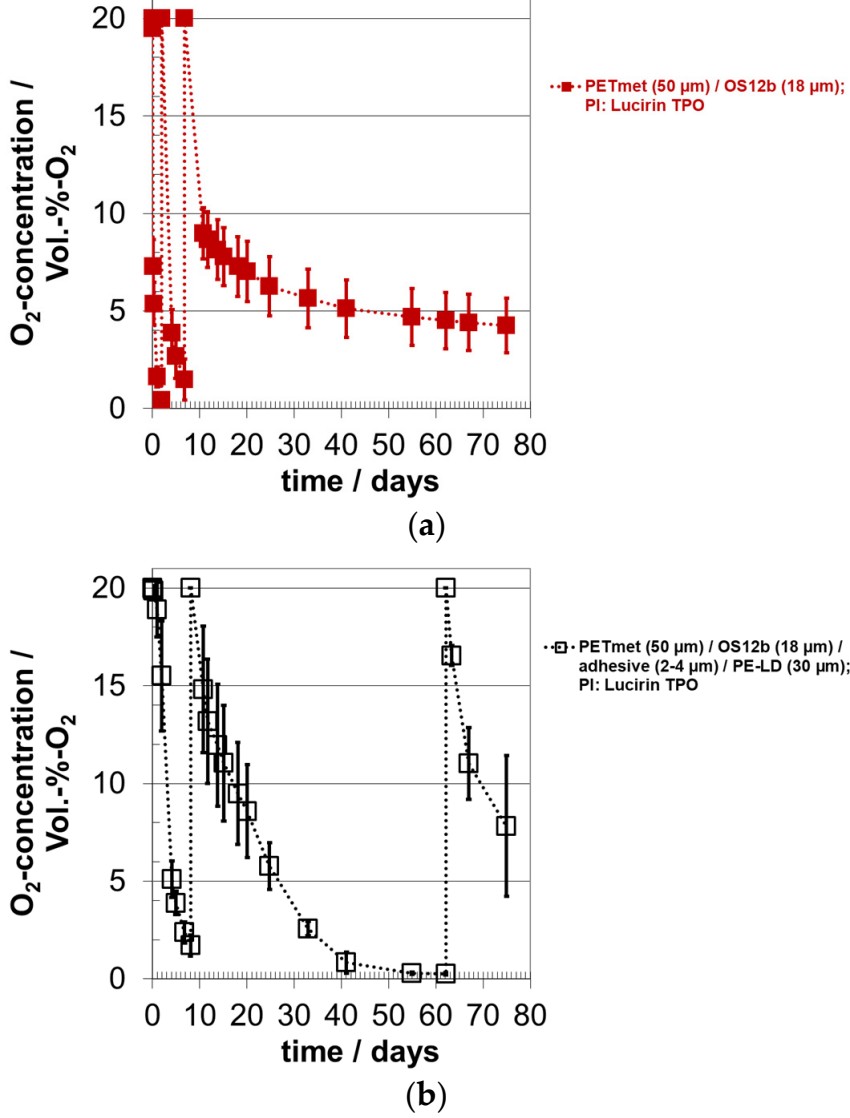

**Figure 7.** *Cont.*

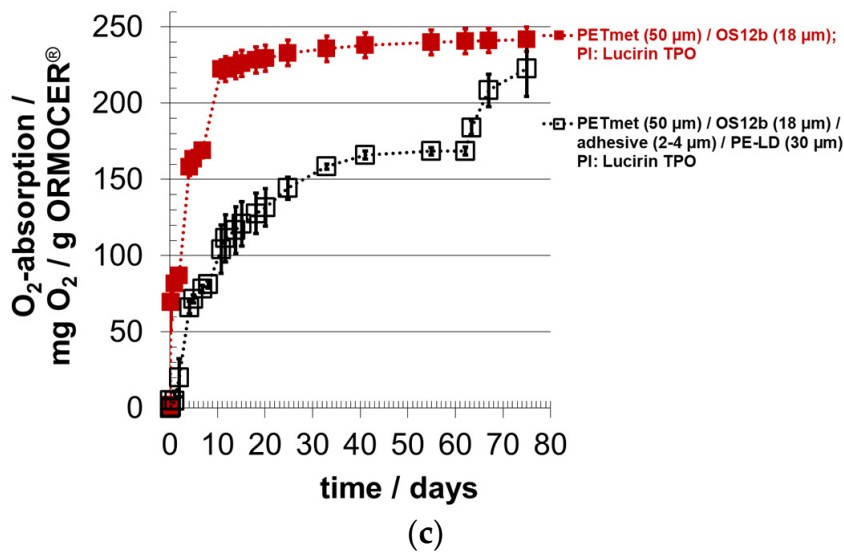

**Figure 7.** Oxygen absorption of oxygen scavenger 'ORMOCER® OS12b' with photo-initiator (PI) Lucirin TPO; curing time: 10 min; OS coating thickness (dry) around $18.2 \pm 3.5$ μm (corresponds to $17.1 \pm 3.3$ g/m$^2$); with and without sealing layer PE-LD; (**a**,**b**) course of the oxygen concentration, (**c**) course of the oxygen absorption.

### 3.1.2. Oxygen Absorption Capacity Compared with Other Oxygen Scavengers

The maximum oxygen absorption of the ORMOCER®-based scavenger determined in this study is much higher than the oxygen absorption capacity of other oxygen scavengers reported in the literature (Table 1). The absorption capacity was twice as high as a similar system tested before and as OSP™, which reacts in a similar way. The high absorption capacity can be explained by the degrees of freedom during the syntheses of the ORMOCER®-based scavengers, which allow a high share of active compound. The other oxygen scavengers must be extruded. To be processable, the concentration of active compound is restricted. With oxygen scavengers for PET, the values refer to the oxygen scavenger additive. It is used in the market at a concentration of <10 wt.% [22,24,42]. An obvious reason is the limitation by food contact material regulations (EU 10/2011). With the iron-based system, iron is 'diluted' in the polymer.

**Table 1.** Oxygen absorption capacities of different oxygen scavengers.

| Oxygen Scavenger | Oxygen Absorption Capacity per Weight Scavenger in mg O$_2$/g Scavenger | Refs. |
|---|---|---|
| In this study: a cyclo-olefin, bonded to a silicate backbone 'ORMOCER®'; applied with wet coating with subsequent drying; activation: UV light | >100 to 240 | - |
| Cyclo-olefin bonded to a silicate backbone 'ORMOCER®', previous study; wet coatings with subsequent drying; activation: UV light | 90 | [32] |
| Ethylene methylacrylate cyclohexenylmethyl acrylate 'OSP™'; separate layer, e.g., in multilayer; activation: UV light | 60 to 100 | [3,20] |
| Iron-based systems ('SHELFPLUS™'); additive for polymer layers of multilayer structures; activation: humidity | 25.4 to 86 | [38,43,44] |
| Metal-catalyzed poly(1,4-butadiene); activation: contact with catalyst during extrusion | 140 | [45] |
| O2Block® (NanoBioMatters S.L., Paterna, Spain); additive for polymers; activation: humidity | >10–25 | [46] |

**Table 1.** *Cont.*

| Oxygen Scavenger | Oxygen Absorption Capacity per Weight Scavenger in mg $O_2$/g Scavenger | Refs. |
|---|---|---|
| Copolyester-based polymer (Amosorb DFC 4020, ColorMatrix Group Inc., Texas, US); additive for PET; activation: contact with catalyst during extrusion | 20–60 | [47] |
| Copolyester-based polymer (Amosorb DFC 4020, Colormatrix Europe, Liverpool, UK); additive for PET; activation: contact with catalyst during extrusion | 43–47 | [48] |
| MXD6 with catalyst; additive for PET; activation: contact with catalyst during extrusion | >77 | [21] |
| 'Oxyclear', polymer with catalyst; additive for PET; activation: contact with catalyst during extrusion | >300 | [49] |

In contrast to the iron-based system, which is activated when in contact with humidity, the ORMOCER®-based scavenger is active in wet conditions, when it is activated by UV light. Activity in wet conditions can be assumed; it was, however, not tested during this study.

### 3.2. Oxygen Partial Pressure in Experimental Packaging

An experimental food packaging sample was tested to evaluate the relevance for food packaging [38]. To absorb the approximately 10 mg oxygen per snack package that can come in to contact with this snack product during its shelf life of 180 days, a scavenger lacquer layer thickness of 2.0 μm is required (corresponds to 1.9 g/m$^2$, calculated from the absorption capacity of the ORMOCER® lacquer with sealing layer; see Section 3.1.1). The layers produced for these tests have thicknesses of 3.2 ± 1.3 μm (corresponds to 3.0 ± 1.2 g/m$^2$) and 18.2 ± 3.5 μm (corresponds to 17.1 ± 3.3 g/m$^2$), and are overdesigned.

In Figure 8, the courses of the oxygen partial pressure in the experimental packaging are shown. Due to their reactivity, the produced films reduced the oxygen partial pressure in the measuring cells (Figure 8). The sealing layer reduced the absorption rate. There was no difference between an ORMOCER® layer thickness of 3.2 ± 1.3 μm and 18.2 ± 3.5 μm. From this, it can be concluded that for the applied layer thickness ranges, a larger layer thickness of the oxygen scavenger does not cause faster oxygen absorption at the beginning. The absorption rate was slightly slower compared to a similar experimental setup with an iron-based oxygen scavenger [38].

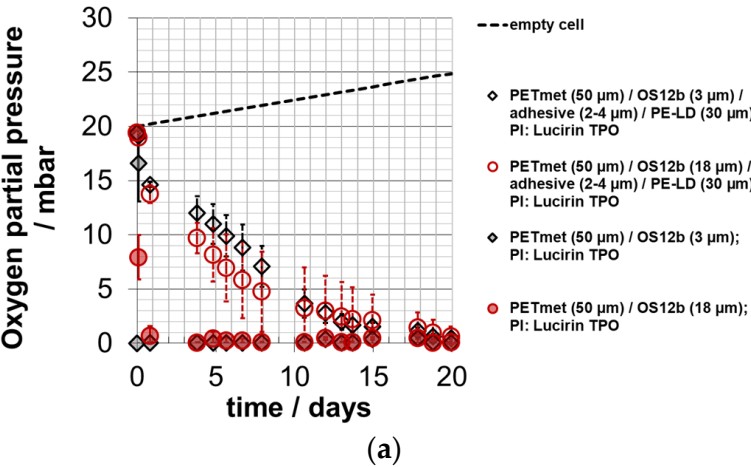

**(a)**

**Figure 8.** *Cont.*

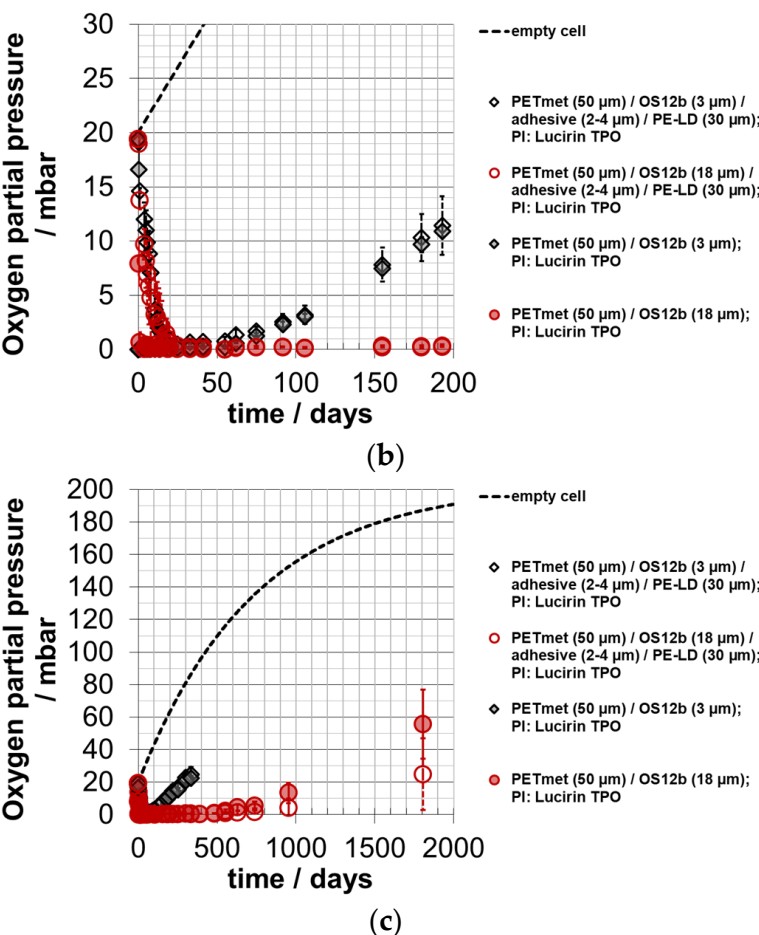

**Figure 8.** Oxygen partial pressure in experimental packaging; absorption of oxygen scavenger 'ORMOCER® OS12b' with photo-initiator (PI) Lucirin TPO; curing time: 10 min; 0% RH; 23 °C; OS coating thickness (dry) around $3.2 \pm 1.3$ μm and $18.2 \pm 3.5$ μm; depiction (**a**) up to day 20; (**b**) up to day 200; (**c**) up to day 2000; empty cell (empty experimental packaging): calculated values.

With a scavenger layer thickness of 18 μm, an oxygen partial pressure of near zero was achieved for almost 500 days. Within the shelf life of 180 days, the oxygen partial pressure of the experimental packaging with 3 μm scavenger layer thickness increased to 10 mbar. In a similar study with an iron-based system, the partial pressure increased during this time to 3 mbar. This is approximately less by a factor of 3 for the 3 μm scavenger layer thickness and more by a factor of 3 for the 18 μm scavenger layer thickness [38]. Therefore, the snack product is—to a certain extent—already deteriorated at the 3 μm scavenger layer thickness due to the lower scavenger layer thickness. However, it is better protected at the 18 μm scavenger layer thickness. Therefore, the oxygen scavenger layer thickness should be higher than 3 μm but not more than 18 μm here.

## 4. Conclusions and Outlook

This study has shown that ORMOCER®-based oxygen scavengers are attractive for food, pharmaceutical and technical packaging. The system is transparent, and it is triggered by UV light shortly before packaging or even through a (UV) transparent packaging. For this reason, the system can be stored for some time in air before activation. The ORMOCER®-based oxygen scavenger has a high absorption capacity and rate. With an absorption capacity of $242 \pm 8$ mg $O_2$/g scavenger, it is one of the most potent oxygen scavengers for packaging-related applications documented in the scientific literature. As a result, the thickness of the active layer can be significantly reduced (to a few micrometers) compared to other commercially available systems. Due to the high reaction rate, the

oxygen concentration in packaging can be reduced to near zero within hours or at least within few days. The system is suitable for roll-to-roll applications and application on flexible substrates, but also on three-dimensional shapes. The system is also attractive for high-barrier films to support the barrier layers.

The ORMOCER®-based oxygen scavenger is not at all or only marginally affected by the two-component polyurethane laminating adhesive used. The non-laminated oxygen scavenger layer absorbed $242 \pm 8$ mg $O_2$/g scavenger and the laminated oxygen scavenger layer absorbed about $223 \pm 18$ mg $O_2$/g scavenger. Therefore, it is possible to cover the scavenger layer with a functional barrier layer that mitigates the potential migration of photo-initiator from the scavenger layer to the packaged products such as food or pharmaceuticals. Hence, the applicability of the scavenger in a multilayer structure with food or pharmaceutical contact is viable.

**Author Contributions:** Conceptualization, S.S. and S.A.-S.; methodology, S.A.-S., S.S. and K.M.; investigation, S.A.-S., S.S. and K.M.; resources, S.A.-S., S.S. and K.M.; data curation, S.A.-S., S.S. and K.M.; writing—original draft preparation, S.S.; writing—review and editing, S.A.-S., S.S., K.M. and F.S.; project administration, S.A.-S., S.S. and K.M.; funding acquisition, S.A.-S., S.S. and K.M. All authors have read and agreed to the published version of the manuscript.

**Funding:** We thank the 'Stiftungsfond UNILEVER', 'Deutsches Stiftungszentrum (DSZ) im Stifterverband für die Deutsche Wissenschaft' (project-number T 022/20557/2010/kg) for funding. Our research was additionally supported by internal funds of the Fraunhofer-Institute for Silicate Research ISC and Fraunhofer Institute for Process Engineering and Packaging IVV.

**Institutional Review Board Statement:** Not applicable.

**Informed Consent Statement:** Not applicable.

**Data Availability Statement:** The data presented in this study are available on request from the corresponding author.

**Acknowledgments:** The authors thank Erika Friedel from Fraunhofer ISC in Würzburg for proofreading.

**Conflicts of Interest:** The authors declare no conflict of interest.

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
