# Peer review of "UV-Activated, Transparent Oxygen Scavenger Coating Based on Inorganic–Organic Hybrid Polymer (ORMOCER®) with High Oxygen Absorption Capacity"

_coatings, doi:10.3390/coatings13020473_

Round 1
Reviewer 1 Report
This work deals with the preparation of a coating material based on inorganic-organic polymers, which has oxygen scavenger properties when it is activated with UV light. In particular, after the synthesis of the scavenger, it was coated on an aluminum foil to work as material for packaging applications. Overall, this work is interesting and it falls within the scope of the journal. However, there are some major revisions that need to be addressed:
1.More information about the scavenger chemical nature and properties should be provided. In the preparation section, authors report about the “addition of a catalyst under ice cooling.” (line 126). What is the catalyst?
2.Being a coating materials that can be used for many applications, such as food industry, the wettability should be investigated. Contact angle measurements can help.
3.What is the effect of this coating on the mechanical properties and thermal stability of the resulting materials? These properties are crucial as well and more efforts should be made in view of them.
4.Please improve the quality of figures, above all fig. 7, as they are very disordered. Maybe authors could delete the lines connecting experimental points.
5.The conclusions should be enriched with some experimental findings.
6.The introduction should be enriched with the most recent findings about coating materials with different purposes (please see: https://doi.org/10.3390/coatings13010195, https://doi.org/10.1016/j.matlet.2022.133257, https://doi.org/10.1016/j.hybadv.2023.100020).
Author Response
Dear Reviewers,
Dear Editor,
Thank you for reading our manuscript and your valuable comments and suggestions. We addressed all comments and amended our document.
Editor and Reviewer comments:
Reviewer 1:
“1. More information about the scavenger chemical nature and properties should be provided. In the preparation section, authors report about the “addition of a catalyst under ice cooling.” (line 126). What is the catalyst?”
Information is amended and added. The catalyst is cobalt di-oleate. This information is written section 2.1
“2.Being a coating materials that can be used for many applications, such as food industry, the wettability should be investigated. Contact angle measurements can help.”
We added this information in section 2.2. Since this property was not in the focus of this study we provide this information in the Materials and Methods section and not in Results and Discussion.
“3.What is the effect of this coating on the mechanical properties and thermal stability of the resulting materials? These properties are crucial as well and more efforts should be made in view of them.”
We added this in information in section 2.2. Since this property was not in the focus if this study we provide this information in the Materials and Methods section and not in Results and Discussion.
“4.Please improve the quality of figures, above all fig. 7, as they are very disordered. Maybe authors could delete the lines connecting experimental points.”
We made of Fig. 7a two figures. For researchers working with oxygen scavengers it is of value to see what oxygen concentrations were reached during the experiments.
“5.The conclusions should be enriched with some experimental findings.”
The conclusions were enriched by specific result.
“6.The introduction should be enriched with the most recent findings about coating materials with different purposes (please see: https://doi.org/10.3390/coatings13010195, https://doi.org/10.1016/j.matlet.2022.133257 , https://doi.org/10.1016/j.hybadv.2023.100020).”
The authors added several recent references. However, no relevant oxygen scavenger coatings based on inorganic-organic structures could be found. Since inorganic-organic polymers and polymers based on sol-gel technique are a very broad field we decided not to cite review articles and other scientific articles that do not fit well within the scope of this manuscript.
Yours Sincerely
Würzburg, Freising, Munich
Sabine Amberg-Schwab, Kajetan Müller, Ferdinand Somorowsky, Sven Sängerlaub
Reviewer 2 Report
The manuscript entitled “UV-activated, Transparent, Oxygen Scavenger Coating based on Inorganic-Organic Hybrid Polymer (ORMOCER®) with High Oxygen Absorption Capacity” could be interesting for the readers. However, the paper needs a significant revision before publication. I have listed a few comments that need to be addressed:
1. I would advise improving the introduction part with up-to-date citations.
2. Adding a recent review on packaging could be useful to improve the literature review.
3. Add more concrete results in the abstract part.
4. what is the novelty of this work that should be clearly discussed at the end of the introduction?
5. Write the full form once when mentioning the first instance.
6. Improve the quality of the images and figures.
7. Add a schematic illustration to show the overall work.
8. Why PET, why not biopolymers as a replacement for synthetic plastic?
9. Table 1 could be more informative.
10. What about the toxicity of the oxygen scavenger organosiloxane that should be added and discussed?
11. Add real-time packaging test using a food system to show the efficiency of the oxygen scavenger.
12. Results are not discussed well, they should be improved carefully with previously published literature.
13. The integration of the results from different parameters should be improved carefully with up-to-date citations.
14. Check the format of the reference, and make it as per the guideline.
15. Conclusion could be better.
16. Also, carefully revise the typos and linguistic errors to make the manuscript error-free.
Author Response
Dear Reviewers,
Dear Editor,
Thank you for reading our manuscript and your valuable comments and suggestions. We addressed all comments and amended our document.
Editor and Reviewer comments:
Reviewer 2:
“1. I would advise improving the introduction part with up-to-date citations.”
We added several up-to-date citations.
“2. Adding a recent review on packaging could be useful to improve the literature review.”
We added several relevant up-to-date citations.
“3. Add more concrete results in the abstract part.”
More results are added.
“4. what is the novelty of this work that should be clearly discussed at the end of the
introduction?”
In 1.3 more information was added. Information about the improvement and optimisation of the scavenger is published. This study presents therefore the results for this optimised oxygen scavenger system.
“5. Write the full form once when mentioning the first instance.”
We checked the whole document an amended it accordingly.
“6. Improve the quality of the images and figures.”
In Figure 7 the quality was improved. The other Figures were checked.
“7. Add a schematic illustration to show the overall work.”
We amended the manuscript. Fig. 1 and Fig. 2 present the system well. We hope these descriptions are sufficient.
“8. Why PET, why not biopolymers as a replacement for synthetic plastic?”
The focus here was to develop a new oxygen scavenger. The used non-biobased polymers such as PE and PET can easily be substituted by bio-PE and bio-PET. To combine the scavenger with other biopolymers is of interest for Future studies.
“9. Table 1 could be more informative.”
We added more information.
“10. What about the toxicity of the oxygen scavenger organosiloxane that should be added and discussed?”
More information was added in the introduction.
“11. Add real-time packaging test using a food system to show the efficiency of the oxygen scavenger.”
We did tests with an experimental packaging as described in 2.6. Real food tests are planned in Future.
“12. Results are not discussed well, they should be improved carefully with previously published literature.”
The authors did an extensive search for suitable references. We added them were suitable. Not many specifically relevant references could be found.
“13. The integration of the results from different parameters should be improved carefully with up-to-date citations.”
The authors did an extensive search for suitable for references. We added them were suitable. Not many specifically relevant references could be found.
“14. Check the format of the reference, and make it as per the guideline.”
We changed the references to ACS-style as recommended by coatings.
“15. Conclusion could be better.”
We added more specific information to the conclusions.
“16. Also, carefully revise the typos and linguistic errors to make the manuscript error-free.”
We checked the document.
Yours Sincerely
Würzburg, Freising, Munich
Sabine Amberg-Schwab, Kajetan Müller, Ferdinand Somorowsky, Sven Sängerlaub
Round 2
Reviewer 1 Report
The authors addressed the suggestions and corrections. The paper can be accepted for publication.
Author Response
Thank you.
Reviewer 2 Report
The review responses are very brief and not adequate. Authors are advised to concisely revise the manuscript as per the comments, especially comments 11, 12, and 13.
Author Response
14.02.2023
Dear Reviewer,
Dear Editor,
Thank you for reading our manuscript and your valuable comments and suggestions. We addressed all comments and amended our document.
Editor and Reviewer comments:
“The review responses are very brief and not adequate. Authors are advised to concisely revise the manuscript as per the comments, especially comments 11, 12, and 13.”
We revised and amended the document. To avoid speculation, we restricted discussions in some parts of our study. We agree with the reviewer that more discussion would be of value and is open. However, we have the impression that more measurements would be required to support further discussion. The important result of developing an oxygen scavenger coating with high absorption capacity is to our understanding of value to the coating community. We plan refined tests in future.
“11. Add real-time packaging test using a food system to show the efficiency of the oxygen scavenger.”
We did tests with an experimental packaging as described in 2.6 and 3.2. Tests with food are resource intensive. Such extensive tests with food were not within the intention of this study. Here, the focus was to gain absorption results for the new oxygen scavenger. Nonetheless, food tests are planned in Future.
“12. Results are not discussed well, they should be improved carefully with previously published literature.”
The authors did an extensive search for suitable references. We added them were suitable. Not many specifically relevant references could be found. The problem here is that the system described in our study is somehow unique. The focus of this study was to gain results for the improved oxygen scavenger. We used a phenomenological approach. We plan in future studies to examine and describe the reactions deeper applying more methods the allow deeper insights.
“13. The integration of the results from different parameters should be improved carefully with up-to-date citations.”
The authors did an extensive search for suitable for references. We added them were suitable. Not many specifically relevant references could be found. As written in 12 the system is unique and therefore specific up-to-date are missing.
Yours Sincerely
Würzburg, Freising, Munich
Sabine Amberg-Schwab, Kajetan Müller, Ferdinand Somorowsky, Sven Sängerlaub